# Large Language Models for Constrained-Based Causal Discovery

**Kai-Hendrik Cohrs**[1], **Emiliano Diaz**[1], **Vasileios Sitokonstantinou**[1],
**Gherardo Varando**[1], **Gustau Camps-Valls**[1]

[1]Image Processing Laboratory (IPL). Universitat de València, Spain. kai.cohrs@uv.es

## Abstract

Causality is essential for understanding complex systems, such as the economy, the brain, and the climate. Constructing causal graphs often relies on either data-driven or expert-driven approaches, both fraught with challenges. The former methods, like the celebrated PC algorithm, face issues with data requirements and assumptions of causal sufficiency, while the latter demand substantial time and expertise. This work explores the capabilities of Large Language Models (LLMs) as an alternative to domain experts for causal graph generation. We frame conditional independence queries as prompts to LLMs and employ the PC algorithm with the answers. The performances of the LLM-based conditional independence oracle on systems with known causal graphs show a high degree of variability. We improve the performance through a proposed statistical-inspired voting schema that allows control over false-positives and false-negatives rates. Finally, we apply the LLM-based PC algorithm to a complex set of variables around food insecurity in the Horn of Africa and find a plausible graph. Inspecting the chain-of-thought argumentation, we occasionally find causal reasoning to justify its answer to a probabilistic query.

## Introduction

Understanding causality is imperative across various disciplines, as it offers critical insights into the mechanisms of complex systems. For example, in the Earth and climate sciences, uncovering the causal relationship between greenhouse gas emissions and global warming has informed international climate agreements and spurred initiatives for renewable energy adoption, aiding in the mitigation of climate change impacts and promoting environmental sustainability (Stips et al. 2016).

Unraveling causality not only enhances our understanding of the underlying processes, but also empowers us to make informed decisions and take proactive measures to address pressing challenges, thus ensuring the well-being and sustainability of our societies and the environment. Investigating causality poses considerable challenges, with constructing causal graphs representing a particularly formidable task. Data-driven causal discovery methods, including prominent techniques like PC (Spirtes, Glymour,

and Scheines 2000) and GES (Chickering 2002), encounter a range of issues. These methodologies rely heavily on copious amounts of data, necessitating complex conditional independence tests that can be particularly challenging, especially when working with diverse and mixed data types. The assumption of causal sufficiency, which presumes that all relevant variables are observed, can lead to erroneous conclusions, especially when unobserved variables act as potential confounders between system variables. Notably, there exist alternative methods such as LPCMCI (Gerhardus and Runge 2020), FCI (Spirtes, Glymour, and Scheines 2000), SVAR-FCI (Malinsky and Spirtes 2018), and GPS (Claassen and Bucur 2022) that do not assume causal sufficiency (Camps-Valls et al. 2023). Nonetheless, missing data and selection bias continue to pose persistent challenges in real-world applications, prompting efforts to develop more resilient causal discovery methods (Camps-Valls et al. 2023).

In addition to data-driven causal discovery methods, another approach for creating causal graphs involves leveraging domain knowledge. However, this process is inherently challenging and time-consuming, demanding substantial expertise and labor (Long et al. 2023b). Experts tasked with constructing causal graphs must possess a deep understanding of the relationships and mechanisms within the system under investigation. This often entails extensive consultations, discussions, and reviews with domain specialists, adding significant time and resource commitments to the process. Furthermore, the complexity of many real-world systems amplifies the difficulty of accurately capturing all relevant causal relationships, leading to potential oversights and inaccuracies in the resulting causal graph. These challenges underscore the necessity for more automated methodologies. In that respect, LLMs could play a key role if they prove to be a reliable source of causal knowledge.

LLMs present a promising knowledge-driven alternative to expert-based graph building or data-driven causal discovery methods. They have shown good performance across a range of language understanding and logical reasoning tasks (Brown et al. 2020; Xu et al. 2023). This could extend to probabilistic and causal reasoning, including interventional and counterfactual scenarios. Whether or not this is the case is the subject of heated debate (Hobbhahn, Lieberum, and Seiler 2022; Willig et al. 2023; Zečević et al. 2023). Despite the inherent complexity of directly asking LLMs to provide

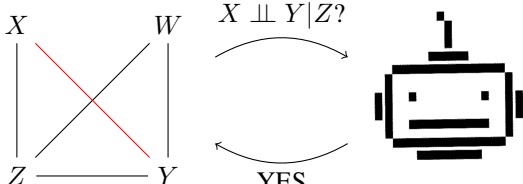

Figure 1: Illustration of the introduced scheme for PC with GPT/LLM. Credits: Little robot face by Antònia Font.

complete causal graphs, given the various levels of reasoning involved and the rich context necessary, different strategies exist to elicit causal graphs from LLMs effectively. These strategies include LLMs to predict causal relations (Kıcıman et al. 2023; Long et al. 2023a; Willig et al. 2023), LLMs as priors for data-driven causal discovery methods (Ban et al. 2023), and LLMs to aid in downstream causal inference tasks by predicting the causal order of variables (Vashishtha et al. 2023).

To marry the traditional and the LLM-based methods, our work proposes *chatPC* as a hybrid approach. Specifically, this work:

- Frames conditional independence queries as prompts to LLMs and employs the PC algorithm with this oracle for causal graph construction.
- Evaluates the performance of LLMs on conditional independence tests across various problems, showing varied performance.
- Introduces a statistical-based approach for aggregating multiple LLM answers, improving performance.
- Examines the graphs predicted by PC with the LLM oracle, finding them to be reasonable.
- Identifies a general tendency for conservative answers from LLMs compared to experts and finds traces of causal reasoning in the model's answers.
- Suggests that the approach could mitigate limitations of traditional methods, offering a promising avenue for automated causal graph construction.

We argue that relying solely on bivariate causal queries inherently overlooks the presence of mediators, consequently limiting the comprehensive understanding of the full causal graph. Additionally, our research explores the extent to which LLMs' queries can effectively substitute data-driven conditional independence tests, considering that PC represents the current state of the art in causal discovery, and under a perfect oracle, it can consistently retrieve the ideal graph up to the Markov equivalence class (Spirtes, Glymour, and Scheines 2000).

## Conditional Independence Queries via LLM

We start by describing and evaluating conditional independence (CI) queries with LLMs. Specifically, we are interested in estimating the validity of conditional independence statements of the type:

*Is X independent of Y given $Z = (Z_1, Z_2, \ldots, Z_k)$?*,

without having access to observations of the involved variables. Instead, we would like to rely on *available or expert* knowledge accessible through LLMs.

For an LLM to be able to answer CI queries, it needs to be presented with some context and additional information related to the variables of interest alongside their description. In particular, we assume that for each *problem*, we have access to the following information:

**variables** names or acronyms plus a short description for each quantity of interest.

**field** the general subject area or expert field related to the problem.

**context** a description of the broad context of the variables under consideration, including relevant details that go beyond general knowledge

## Prompting for conditional independence testing

While the approach we propose in this work could be implemented with any LLM trained to follow instructions, we employ gpt-3.5-turbo from OpenAI[1]. For a CI statement ($X \perp\!\!\!\perp Y|Z$), we consider a simple prompt that combines the field and context information and a description of the involved variables ($X, Y$, and eventually $Z$) with a general instruction and response template as follows (see the Appendix for a detailed specification of the used prompt):

```
Persona specification
Instructions
Context
Variables description
CI Statement question
Response template
```

The persona is based on the field variable and primes the LLM to produce reasoning appropriate to the area under study. To improve the answers, we apply chain-of-thought prompting following Wei et al. (2023), enabling us to gain insights into the model's reasoning and inspect if it is causally inspired. Further, we ask the model to provide uncertainty about its best guess along the lines of Tian et al. (2023).

## Testing

Various strategies could be envisaged to perform a "Hypothesis test" for a conditional independence statement with LLMs. A naive option consists of asking the LLM a single question with the prompt described in the previous section and decide that a statement is valid (i.e., the variables are indeed independent) if the answer is YES and otherwise, it is NO. The main problem with this approach is that since LLMs are probabilistic, a single answer from an LLM does not need to correspond to its mode (the most likely answer) or could fail to respect the required response template and answer, for instance, UNCERTAIN instead of YES/NO. Instead, we ask for an independent batch (size $n$) of answers, parse the obtained answers (YES or NO) together with the reported uncertainties, and finally output either an

---

[1]https://platform.openai.com/docs/models/gpt-3-5

answer based on simple voting or weighted voting where the weights are the reported probabilities.

Alternatively, we implement a "statistical approach" where we actually produce $p$-values for the null hypothesis $p_{no} \geq p_{yes}$ (or alternatively $p_{no} \leq p_{yes}$) where $p_{no}, p_{yes}$ are the proportion of NO and YES answers over the total requested batch $n$. The constructed test is based on the idea that we want to test if the probability of obtaining the answer NO is significantly higher than that of obtaining the answer YES and vice-versa. If we find the difference non-significant in light of the obtained responses, we opt for the null hypothesis. A final decision can then be obtained by setting a significance level $\alpha$ and reject the chosen null hypothesis if the p-value is less or equal to $\alpha$ (we will employ $\alpha = 0.05$ in the experiments). This last strategy has the advantage of considering the random variability of the answers and could offer a principled way of controlling the false positive rate. The user could then specify, for a particular problem, which of the two null hypotheses they would like to employ (either $p_{no} \geq p_{yes}$ or $p_{yes} \geq p_{no}$), which in turn implies a different false-positive control (considering either NO or YES as positive).

## Evaluation

We evaluate the performance for CI testing on various problems defined in the BNLearn repository (Scutari and Denis 2014), the spurious correlation website (Vigen 2023), and a classical problem on reconstructing protein-signaling networks (Sachs et al. 2005):

- cancer Simple causal graph involving four factors influencing the probability of cancer (Korb and Nicholson 2010).

- burglary A modification of the classical earthquake example in (Korb and Nicholson 2010; Scutari and Denis 2014)

- asia Causal graph of eight factors linked to respiratory problems (Lauritzen and Spiegelhalter 1988).

- sachs: Causal graph among 11 phosphorylated proteins and phospholipids in single-cell data (Sachs et al. 2005).

- spurious: Famous examples of pairs of variables that are spuriously correlated, obtained from the spurious-correlation website (Vigen 2023).

For the small problems (burglary and cancer) we evaluate all possible CI statements over 5 variables with both permutations of $X$ and $Y$ (160 statements per problem). For sachs and asia we evaluate all CI statements (with $X$-$Y$ permutations) up to conditioning sets of a certain size (0 and 1, respectively) plus 100 random valid CI statements. Additionally for sachs, we evaluate also 100 randomly chosen statements with a conditioning set of size less or equal to three. Lastly, for spurious we evaluate the marginal independence statements $X \perp\!\!\!\perp Y$ for all pairs of variables. For all the experiments, we obtain the answer to the CI queries by aggregating, as described previously, $n = 20$ independent batched responses from the LLM.

Table 1: Performance of LLM-based conditional independence tests with different voting procedures and metrics (accuracy, precision, recall, and F1 score.

| Dataset | Prediction method | Acc. | Prec. | Rec. | F1 |
|---------|-------------------|------|-------|------|-----|
| burglary | voting | 0.54 | 0.26 | 0.36 | 0.30 |
| | weighted voting | 0.55 | 0.29 | 0.45 | 0.36 |
| | stat. Test ($H_0$: $\perp\!\!\!\perp$) | 0.53 | 0.33 | **0.73** | **0.46** |
| | stat. Test ($H_0$: $\not\!\perp\!\!\!\perp$) | **0.69** | **0.38** | 0.23 | 0.29 |
| | NO | 0.73 | 0.00 | 0.00 | 0.00 |
| cancer | voting | 0.88 | 0.00 | 0.00 | 0.00 |
| | weighted voting | 0.88 | 0.00 | 0.00 | 0.00 |
| | stat. Test ($H_0$: $\perp\!\!\!\perp$) | 0.88 | 0.00 | 0.00 | 0.00 |
| | stat. Test ($H_0$: $\not\!\perp\!\!\!\perp$) | 0.88 | 0.00 | 0.00 | 0.00 |
| | NO | 0.88 | 0.00 | 0.00 | 0.00 |
| asia | voting | 0.79 | 0.10 | 0.18 | 0.12 |
| | weighted voting | 0.78 | 0.09 | 0.18 | 0.12 |
| | stat. Test ($H_0$: $\perp\!\!\!\perp$) | 0.68 | **0.15** | **0.59** | **0.24** |
| | stat. Test ($H_0$: $\not\!\perp\!\!\!\perp$) | **0.86** | 0.08 | 0.06 | 0.07 |
| | NO | 0.91 | 0.00 | 0.00 | 0.00 |
| sachs | voting | 0.58 | 0.48 | 0.87 | **0.62** |
| | weighted voting | 0.57 | 0.47 | 0.86 | 0.61 |
| | stat. Test ($H_0$: $\perp\!\!\!\perp$) | 0.52 | 0.45 | **0.99** | **0.62** |
| | stat. Test ($H_0$: $\not\!\perp\!\!\!\perp$) | **0.61** | **0.50** | 0.46 | 0.48 |
| | NO | 0.61 | 0.00 | 0.00 | 0.00 |

**Permutation consistency** The result of a conditional independence test should not depend on the order of variables, i.e., it should be commutative in $X$ and $Y$ (given $Z$). As a first sanity check, we checked the consistency of the responses with respect to the change of order of $X$ and $Y$ (see Figure 4 in the Appendix).

In over 80% of the cases, the majority votes resulted in the same response, while roughly 13% of the statements disagreed. In the remaining cases, at least one direction resulted in a tie. Given the variance in the generated responses and the majority vote over merely 20 queries, some mismatch is expected. Overall, the LLM seems sufficiently consistent in its responses under change of order. Nevertheless, we propose to aggregate the results of the queries in both directions to obtain results invariant to the order of the two involved variables.

**Performance of CIT** Table 1 summarizes the evaluations of the conditional independence queries over the different problems. We compute the proposed approaches' standard classification metrics (accuracy, precision, recall, and F1 scores). We use the test that always yields NO (meaning NO independence) as a reference point. It reflects the density of the graph (the higher its accuracy, the denser) as it corresponds to assuming a fully connected graph. The results show varying performance over the different causal graphs. In terms of accuracy, all prediction methods underperform with respect to the constant NO baseline due to the density of the graphs. The statistical test ($H_0$: $\not\!\perp\!\!\!\perp$) approach, which favors dependence, comes closest to this and only loses a few percent points in all cases. For burglary and sachs,

it has the highest precision. In Recall and F1 score, the other statistical test ($H_0$: ⊥⊥) partially outperforms the other methods. As independence is usually the underrepresented class, it usually improves in recall and F1 score. Weighted voting usually helps to steer responses that ended in a tie in favour of the one where the models were more certain but did not change the results substantially compared to the normal vote. Statistical-based approaches have a more principled way of working with cases where the vote is not clear enough and allow the deployer to choose how to control for false positive or false negative rates. Even though the LLM did not excel in this task, the responses for the two statistical-based approaches may serve as additional baselines. In the case of the cancer problem, the response of the LLM corresponded to the NO baseline, as it consistently suggested that there could still be some relationship between the variables. As for health data, there are always many confounders; this seems to be a conservative or safe decision.

**Inquiring spurious correlations** To investigate this further, we went to the other extreme and asked for statistical independence between variables taken from the spurious correlation website (see Table 2 in the Appendix). Here, instead, the model almost always chose independence. At the same time, revenue-CS, spending-suicides, and chicken-oil showed the highest disagreement in the responses. For the statistical-based ($H_0$: ⊥̸⊥), this even leads to a rejection in the case of revenue-CS. Also, in this case, the model often seems to take the conservative answer: correlation by chance, i.e., YES to independence. On the other side, correlation, according to Reichenbach's principle, implies either direct causation or a common cause. We examined the responses of the LLMs to the queries of the above-mentioned pairs. For the pair spending-suicides, for instance, in some answers, it reasons about factors that influence both variables and gives reasonable examples of confounders such as diverting resources or progress in mental health research (see Appendix Response example A for the full response). The mentioned response shows causal reasoning when the model was not directly requested. And it goes beyond the required reasoning for directing edges as in previous works. We find, however, that the model does not consistently take this path in the answer. In some cases, instead, it reasons about the impacts on probability when fixing the conditioning set and makes false statements about determining conditional independence (see Appendix Response example B for the full response). This shows that more work must be done to steer LLMs towards reliable and coherent causal reasoning.

## Application to Causal Discovery

We propose to couple the conditional independence oracle or testing, introduced in the previous section, with the PC-algorithm (Colombo and Maathuis 2014) for recovery of the Markov equivalence class of a causal graph. The PC algorithm starts from a fully connected skeleton (undirected graph) among the considered variables and iteratively removes edges between variables $X$ and $Y$ when it finds a conditioning set $Z$ such that $X$ is independent of $Y$ given

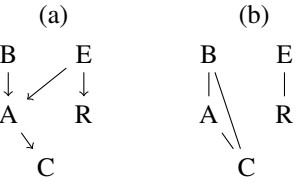

Figure 2: Assumed true graph (a) and skeleton recovered (b) with the proposed chatPC approach for the burglary problem. Variables: Burglary in progress (B); earthquake (E); radio station announcing earthquake (R); alarm ringing (A); security company calling (C).

$Z$. After the so-called skeleton phase, v-structures are identified through specific conditional independence testing, and finally, a set of orientation rules are applied (Meek 1995). Hence, to implement chatPC, we plug in the LLM-based conditional independence testing in an available PC implementation (Atienza, Bielza, and Larrañaga 2022).

## Causal graphs from the examples

Since for the cancer problem, all CI statements are considered to be false (see Table 1), chatPC would retrieve a complete skeleton. The graph obtained in the burglary problem (with the stat. $H_0 = ⊥⊥$ strategy) is depicted in Figure 2 together with the assumed ground truth, while the estimated graph in the asia and sachs problems are reported in the Appendix.

In all cases, the obtained causal graphs have some similarities with the *ground truth* graphs. Given the imperfect results in the CI statements, recovering the true causal graph is impossible. Nevertheless, we can observe that some interesting patterns have been uncovered (e.g. the Plcg-PIP2-PIP3 and Raf-Mek-Erk relationships in sachs) and an almost correct skeleton in the burglary problem has been estimated.

## Exploring the uncertain: Food insecurity in Africa

As an exploratory real-world example, we try to extract the causal graph among 8 variables related to food security in Somalia using chatPC. The Horn of Africa has seen a troubling increase in food insecurity, impacting 6.5 million people in 2022. Prolonged dry spells, along with factors like hydrological conditions, limited food production, market access issues, inadequate humanitarian aid, conflicts, and displacement, contribute to the complex challenges of households in Somalia (Cerdà-Bautista et al. 2023).

The output graph obtained leveraging information of CI statements up to conditioning sets of size 1 is shown in Figure 3. Firstly, we note that the obtained graph is not fully connected. We observe that all obtained arcs and the estimated causal directions are not inconsistent with domain knowledge and common sense (Cerdà-Bautista et al. 2023). Specifically, we can see the following causal relationships, which agree with the dynamics of agropastoralist households in drought displacement situations, as reported by the Internal Displacement Monitoring Center (iDMC) (Internal

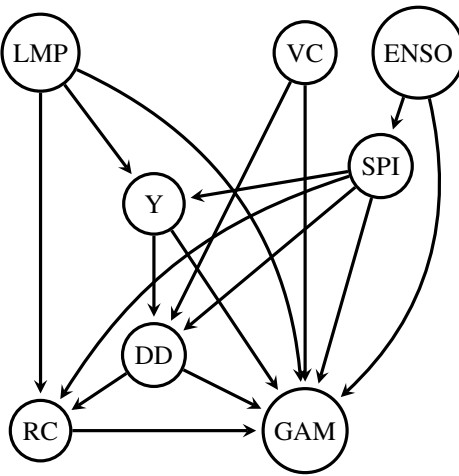

Figure 3: Causal graph obtained from chatPC (`gpt-3.5-turbo`) with the statistical strategy ($H_0 = \perp\!\!\!\perp$) and using the information of conditional independence tests up to conditioning set of size 1. Variables: El Niño Southern Oscillation (ENSO); Standardized Precipitation Index (SPI); Recorded fatalities due to violent conflicts (VC); Local market prices (LMP); Sorghum yield production (Y); Drought-induced internal displacement (DD); Number of individuals that received cash from humanitarian aid (RC); Global Acute Malnutrition (GAM).

Displacement Monitoring Centre (iDMC) 2020); i.e., Violent Conflict (VC) - Drought Displacement (DD), Standardized Precipitation Index (SPI) - Sorghum Yield Production (Y), Local Market Prices (LMP) - Global Acute Malnutrition (GAM).

## Conclusions

Our work contributes to the existing literature by probing an alternative to data-driven PC, leveraging the capabilities of LLMs for PC when data is limited or unavailable. Building a reliable knowledge-based conditional independence *oracle* could either provide a prior to constrain its data-driven counterpart or even deliver a more reliable substitute for data-driven methods. Our analysis attempts to shed light on where we stand in this endeavor. We found that LLM sometimes conjectures about hidden confounders, showing that they use causal reasoning to tackle this primarily statistical task. This, however, is neither done consistently nor always successfully. The varying performance over different tasks showed that more effort is needed to steer the models to more efficient causal reasoning. We proved that employing an aggregating mechanism framed as a statistical test leads to improved performance and effective control over false positive and negative rates. The causal graphs predicted by the PC algorithm with LLM-based conditional independence tests appear reasonable. While not infallible, the method demonstrates potential in capturing meaningful causal relationships, offering a promising avenue for au-

tomated causal graph construction. Finally, we found that LLMs generally tend toward conservative answers, contrasting with the often bolder responses from human experts. Understanding and addressing the cautious nature of LLM reasoning is crucial for refining the accuracy and reliability of the generated causal graphs.

In the future, we will explore the combination of data-driven and language-driven causality, where relying on CIT estimates in PC schemes constitutes a sound framework to improve consistency and robustness.

## Acknowledgements

This work received support from the European Research Council (ERC) under the ERC Synergy Grant USMILE (grant agreement 855187), the GVA PROMETEO AI4CS project on 'AI for complex systems' (2022-2026) with CIPROM/2021/056, and the 'Causal4Africa' project part of the Microsoft Climate Research Initiative.

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

## Details on the Prompt

Here, we describe the details of the prompt used. In the following prompt scheme, curly brackets {} denotes elements that are replaced by corresponding elements from the problem description and the specific CI statement which is been queried. In particular {field} and {context} are replaced by their values for the problem; and {x}, {y} and {z} are the name of the variables involved in the CI ($X \perp\!\!\!\perp Y | Z$). If $Z = \emptyset$ the whole given {z} is dropped from the prompt.

```
1
2  system: You are a helpful expert in {
       field} and willing to answer
       questions.
3
4  system: You will be asked to provide
       your best guess and your uncertainty
       on the statistical independence
       between two variables potentially
       conditioned on a set of variables.
       Your answer should not be based on
       data or observations but on available
        knowledge. Even when unsure or
       uncertain, provide your best guess (
       YES or NO) and the probability that
       your guess is correct. Answer only in
        the required format.
5
6  user: {context} Consider the following
       variables:
7  {variables list and description}
8  is {x} independent of {y} given {z}?
9
10 system: Work out the answer in a step-by
       -step way to be as sure as possible
       that you have the right answer. After
        explaining your reasoning, provide
       the answer in the following form: [<
```

```
      ANSWER> (<PROBABILITY>)] where ANSWER
        is either YES or NO and PROBABILITY
        is a percentage between 0\% and
        100\%.
11  YES stands for "{x} is independent of {y
        } given {z}"  and NO stands for "{x}
         is not independent of {y} given {z}".
12
13  For example [NO (50%)] or [YES (50%)].
```

## Permutation consistency

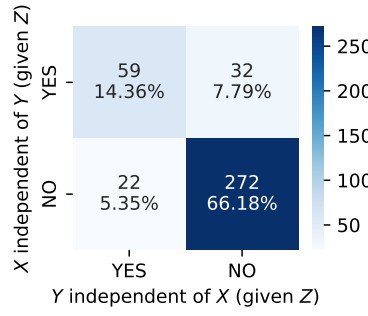

Accuracy=0.808

Figure 4: Confusion matrix of the model's responses to queries with changing order of $X$ and $Y$. Uncertain outcomes in case of a tie in majority voting are hidden.

## Response example A

```
1  Based on the given information, it is
       difficult to determine the
       statistical independence between
       spending on science, space, and
       technology and suicides by hanging,
       strangulation, and suffocation. There
        may be multiple factors that
       influence both variables, such as
       socioeconomic status, mental health,
       and government policies.
2
3  However, contrary to what might be
       expected, there is a possibility that
        spending on science, space, and
       technology could have an indirect
       effect on suicides. Investments in
       science and technology may lead to
       advancements in healthcare, mental
       health research, and improvements in
       overall quality of life, potentially
       reducing suicide rates. On the other
       hand, excessive spending in these
       areas may divert resources from other
        societal needs, which could
       indirectly increase psychosocial
       stressors contributing to suicide
       rates.
4
5  Considering the lack of concrete
       evidence and the potential indirect
```

```
       pathways, we can only make an
       educated guess.
6
7  My best guess would be that spending is
       not completely independent of
       suicides, and there might be some
       association between the two variables
       . However, the strength and direction
        of this association are uncertain.
8
9  [NO (60%)]
```

## Response example B

```
1  To determine if E is independent of B
       given [''C''], we need to assess if
       knowledge of C provides any
       additional information about the
       relationship between E and B.
2
3  Assuming that the alarm system is
       designed to detect burglaries
       accurately but is not specifically
       designed to detect earthquakes, the
       occurrence of an earthquake should
       not affect the probability of a
       burglary in progress. Therefore, the
       variables E and B should be
       statistically independent.
4
5  However, the alarm company will inform
       the owner once the alarm system is
       triggered. The decision to call the
       owner is based on whether the alarm
       company received a signal or not (
       variable C). This introduces a
       conditional relationship between E, B
       , and C.
6
7  Since C is conditioned on B, the
       occurrence of an earthquake (E) may
       impact the probability of the alarm
       system being triggered (C) when a
       burglary is in progress (B), as an
       earthquake could potentially set off
       the alarm system accidentally. Thus,
       the variables E and B may no longer
       be independent given the knowledge of
        C.
8
9  Therefore, E is likely not independent
       of B given [''C''].
10
11  [NO (90%)]
```

## Spurious Correlations

We report in Table 2 the responses obtained for the `spurious` problem over 15 marginal independence statements for the corresponding pairs of spuriously associated variables. We report the decisions obtained with the voting and the two statistical approaches; moreover, the number of NO and YES answers among the $n = 20$ batched response are reported.

Table 2: Predictions for variable pairs of the spurious correlations dataset.

| Variable Name | Description | voting NO - YES | stat. Test ($H_0 : \perp\!\!\!\perp$) | Test ($H_0 : \not\perp\!\!\!\perp$) |
|---|---|---|---|---|
| spending suicides | US spending on science, space, and technology
Suicides by hanging, strangulation and suffocation | YES
4 - 11 | YES | YES |
| pool cage | number of people who drowned by falling into a pool per year
number of films Nicolas Cage appeared in per year | YES
0 - 19 | YES | YES |
| cheese bed | per capita cheese consumption
number of people who died by becoming tangled in their bedsheet | YES
0 - 19 | YES | YES |
| divorce margarine | divorce rate in Maine
per capita consumption of margarine | YES
0 - 12 | YES | YES |
| age murder | age of Miss America
number of people murdered by steam, hot vapors and hot objects | YES
3 - 15 | YES | YES |
| revenue CS | total revenue generated by arcades
computer science doctorates awarded in the US | YES
5 - 9 | YES | NO |
| launches Soc | worldwide non-commercial space launches
sociology doctorates awarded (US) | YES
3 - 15 | YES | YES |
| mozzarella engineering | per capita consumption of mozzarella cheese
civil engineering doctorates awarded | YES
3 - 15 | YES | YES |
| boat Kentucky | people who drowned after falling out of a fishing boat
marriage rate in Kentucky | YES
0 - 18 | YES | YES |
| Norway railway | US crude oil imports from Norway
drivers killed in collision with railway train | YES
4 - 15 | YES | YES |
| chicken oil | per capita consumption of chicken
US crude oil imports | YES
5 - 14 | YES | YES |
| swimming-pool power | number people who drowned while in a swimming-pool
power generated by US nuclear power plants | YES
3 - 16 | YES | YES |
| cars crashing | Japanese passenger cars sold in the US
Suicides by crashing of motor vehicle | YES
3 - 12 | YES | YES |
| spelling spiders | letters in winning word of Scripps National Spelling Bee
number of people killed by venomous spiders | YES
0 - 20 | YES | YES |
| maths uranium | math doctorates awarded
uranium stored at US nuclear power plants | YES
2 - 14 | YES | YES |

## Estimated Graphs

In Figure 5 and 6, we report the estimated graphs for the `asia` and `sachs` problems, respectively. For `asia` only information on the CI statements up to conditioning sets of size 1 are used, while for `sachs` only marginal independence statements are queried. Higher-order CI statements, that is CI with larger size of conditioning sets, are not used for pruning the skeleton (meaning the oracle always returns NO).

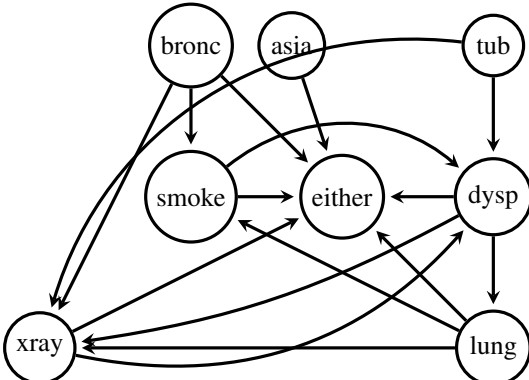

Figure 5: Estimated graph for the `asia` problem, obtained with chatPC (stat. $H_0 = \perp\!\!\!\perp$). Only information on CI statements up to conditioning sets of size 1 is used.

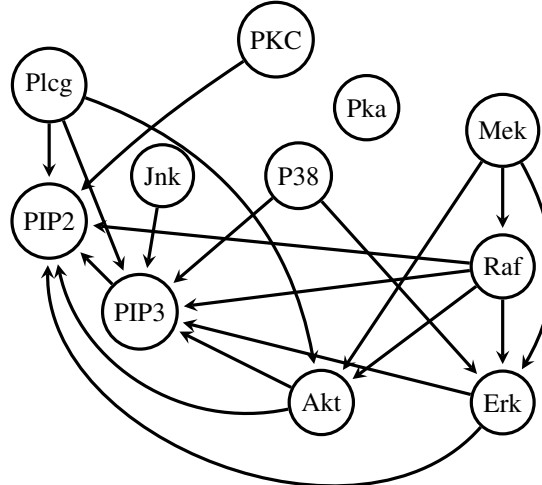

Figure 6: Causal graph for the `sachs` problem, obtained with chatpC (stat. $H_0 = \perp\!\!\!\perp$). Only information on marginal independence statements is used.