# OpenReview forum: "Large Language Models for Constrained-Based Causal Discovery"
_AAAI.org/2024/Workshop/LLM-CP — LLM-CP @ AAAI 2024 Oral_

### Official Review · Reviewer_WNai · 2023-12-01
**The authors present an interesting idea to use LLMs as a conditional independence (CI) tester and evaluate the quality of such a tester. They discuss the majority of relevant related work, do a good job in presenting the idea and their approach how LLMs can be used as a CI tester and provide an empirical evaluation showing that their approach works in some cases.**

**Rating:** 2
**Confidence:** 2

**Review:**

**Summary**

This paper explores how LLLMs can be used as conditional independence (CI) testers leveraging the vast amount of knowledge encoded in LLMs, hence serving as an alternative approach to expert knowledge. The authors integrate their LLM based CI tester in PC to construct causal graphs based on various well known datasets and show that the performance of PC using such a CI tester is often reasonable.

**Strengths**
- all relevant related work is sufficiently discussed
- the authors do a good job describing their method and make it more robust by using hypothesis tests
- empirical evidence is provided showing that the presented approach can work under appropriate circumstances
- interesting and important real world application

**Weaknesses**
- missing comparison to classical methods such as PC and GES in empirical evaluation. Such a comparison would boost the paper quality.
- minor: "The output graph obtained leveraging information of CI statements up to conditioning sets of size 1 is shown in Figure 3." I think this should be "not fully connected".
- minor:  "considering that PC represents the current state of the art in causal discovery [...]". I think methods like FCI (or more recent extensions of it) should be considered to be state of the art since they weaken the assumption of causal sufficiency

**Questions that remained open to me**
- which hypothesis test did you use?
- how did you orient the non-collider edges in the real world application (e.g. $DD \leftarrow VC \rightarrow GAM$)? Seems to be unlikely that PC was able to recover **all** edge directions, did you use causal explanations provided by the LLM?
- what happens if we have contradictions in the problem statement or contradictions in the LLMs answers? Can they be resolved by chain of thoughts?

---

### Official Review · Reviewer_eS6Z · 2023-12-05
**Approach lacks formalization in the statistical testing, raising concerns about result interpretation and contributions**

**Rating:** 1
**Confidence:** 3

**Review:**

The paper explores an intriguing concept: employing large language models (LLMs) as a means to unveil conditional independencies for causal discovery. However, the authors' approach in framing the LLM as a hypothesis test for conditional independence lacks formalization.

Their method involves querying the LLM multiple times with a specific conditional independence question, assuming the null hypothesis is independence. If the proportion of negative responses (favoring dependence) exceeds the proportion of positive responses (favoring independence), they reject the null hypothesis. The authors, however, refer to the use of a p-value and a significance level of 0.05 in their experimental results, but the computation of this p-value remains unexplained. This omission impedes the interpretation of results and the assessment of the paper's contributions.

It's worth noting that if the p-value is derived solely from proportions, it deviates from the conventional definition of a p-value, which is not the probability of the null hypothesis itself. A Bayesian formalization of this test might offer a more appropriate statistical framework.

In essence, while the idea of leveraging LLMs for causal discovery is compelling, the paper's methodology and lack of formal grounding in statistical testing raise concerns about the reliability and interpretability of the findings. A more robust approach, possibly rooted in Bayesian methods, could enhance the credibility and impact of the research.

---

### Official Review · Reviewer_Hrcz · 2023-12-07
**The authors propose using LLMs to evaluate candidate graphs in causal discovery. This serves as an alternative to domain knowledge and has the potential to speed up causal discovery. The proposed approach, chatPC, implements this idea using the PC algorithm. In my opinion, this idea has merit and the initial results are insightful. Although more work is required to further develop and test this idea, I believe that the work would be a valuable addition to the workshop.**

**Rating:** 3
**Confidence:** 2

**Review:**

There are several points that I would like to point out as possible areas for future work.

The first results show that conditional independence tests are somewhat accurate and that LLMs are fairly consistent. Robustness and reliability are found to be lacking, which is an intuitive finding for LLMs in my opinion. Unfortunately, this means that the resulting causal graph cannot be trusted, which is a major issue for causal discovery. How could we leverage LLMs without sacrificing reliability of the causal discovery algorithm? For example, it could be more interesting to guide the causal discovery algorithm without deriving hard constraints from the LLM (e.g., based on probabilities/soft constraints). In this case, the potenetial benefit would be computational efficiency.

Different prompts could have different results. Therefore, I wonder how much these results depend on the specific prompt used in your work, as well as the specific LLM. Nevertheless, I understand that the authors present an initial study and will likely address these points in the future.

The results are especially poor for the cancer data. Presumably, the LLM will be better for some domains and worse for others. A discussion on this would be insightful (e.g., how can we know a priori which domains are a viable candidate). Similarly, fine-tuning could probably be beneficial?

---

### Meta-Review · Area_Chair_fiYe · 2023-12-13

**Recommendation:** 2
**Confidence:** 3

**Metareview:**

While I agree with reviewer 2 that multiple queries are not sufficient, nonethless, the paper can generate interesting discussions. The authors could benefit from the community and hence I recommend acceptance.

---

### Decision · Program_Chairs · 2023-12-14

**Decision:**

Accept (Oral)

**Comment:**

Thank you for submitting your work to the LLM-CP workshop @ AAAI 2024. See you in Vancouver!